# Use of MALDI-TOF MS to Discriminate between Aflatoxin B1-Producing and Non-Producing Strains of *Aspergillus flavus*

**DOI:** 10.3390/molecules27227861

**Published:** 2022-11-14

**Authors:** Lukas Hleba, Miroslava Hlebova, Anton Kovacik, Jana Petrova, Zuzana Maskova, Juraj Cubon, Peter Massanyi

**Affiliations:** 1Faculty of Biotechnology and Food Sciences, Institute of Biotechnology, Slovak University of Agriculture in Nitra, Tr. Andreja Hlinku 2, 949 76 Nitra, Slovakia; 2Department of Biology, Faculty of Natural Sciences, University of Ss. Cyril and Methodius, Nám. J. Herdu 2, 917 01 Trnava, Slovakia; 3Faculty of Biotechnology and Food Sciences, Institute of Applied Biology, Slovak University of Agriculture in Nitra, Tr. Andreja Hlinku 2, 949 76 Nitra, Slovakia; 4Faculty of Biotechnology and Food Sciences, Institute of Food Sciences, Slovak University of Agriculture in Nitra, Tr. Andreja Hlinku 2, 949 76 Nitra, Slovakia

**Keywords:** *Aspergillus flavus*, aflatoxin B_1_, MALDI-TOF MS, discrimination

## Abstract

Aflatoxin B_1_ (AFB_1_) is one of the most toxic mycotoxins. One of the producers of AFB_1_ is *Aspergillus flavus*. Therefore, its rapid identification plays a key role in various sectors of the food and feed industry. MALDI-TOF mass spectrometry is one of the fastest and most accurate methods today. Therefore, the aim of this research was to develop the rapid identification of producing and non-producing strains of *A. flavus* based on the entire mass spectrum. To accomplish the main goal a different confirmatory MALDI-TOF MS and TLC procedures such as direct AFB_1_ identification by scraping from TLC plates, *A. flavus* mycelium, nutrient media around *A. flavus* growth, and finally direct AFB_1_ identification from infected wheat and barley grains had to be conducted. In this experiment, MALDI-TOF mass spectrometry with various modifications was the main supporting technology. All confirmatory methods confirmed the presence of AFB_1_ in the samples of aflatoxin-producing strains of *A. flavus* and vice versa; AFB_1_ was not detected in the case of non-producing strains. Entire mass spectra (from 2 to 20 kDa) of aflatoxin-producing and non-producing *A. flavus* strains were collected, statistically analyzed and clustered. An in-depth analysis of the obtained entire mass spectra showed differences between AFB_1_-producing and non-producing strains of *A. flavus*. Statistical and cluster analysis divided AFB_1_-producing and non-producing strains of *A. flavus* into two monasteries. The results indicate that it is possible to distinguish between AFB_1_ producers and non-producers by comparing the entire mass spectra using MALDI-TOF MS. Finally, we demonstrated that if there are established local AFB_1_-producing and non-producing strains of *A. flavus,* the entire mass spectrum database identification of aflatoxigenic *A. flavus* strains can be even faster and cheaper, without the need to identify the toxin itself.

## 1. Introduction

Mycotoxins are produced by microscopic filamentous fungi as secondary metabolites. They are characterized by several negative properties such as teratogenicity, mutagenicity, or carcinogenicity [1,2,3]. Their danger mainly lies in the fact that they are very often produced by toxigenic fungi in food or feed and thus can threaten the health of people or animals [4]. Aflatoxins are considered to be the most toxic mycotoxins, especially AFB_1_ is the most potent carcinogenic mycotoxin among other aflatoxins. They can cause diseases named aflatoxicosis which may be acute (which results in death) or chronic (which results in cancer, or other less harmful pathological conditions). Aflatoxins are effective even at very low concentrations [5]. These mycotoxins are produced by microscopic filamentous fungi of the genus *Aspergillus*, most often by the species *A. flavus*, *A. parasiticus*, *A. nomius* [6] but also by other newly discovered producers such as *A. texensis* [7], *A. bombycis*, *A. ochraceoroseus*, *A. nomius*, and *A. pseudotamarii* [8,9]. From the point of view of consumer safety and health protection, prevention against toxicogenic fungi and their mycotoxins is essential, and therefore, various methods of quick and easy detection are being sought [10,11]. There are several ways to detect mycotoxins or their producers (HPLC, LC-MS, ELISA), but many of these analytical methods require highly skilled workers and costly equipment and the time of testing is an equally important parameter. Therefore, it is necessary to find a fast and cheap way to detect mycotoxin producers as well as the mycotoxins themselves in food [12]. One of the possibilities for the rapid detection of the presence of toxigenic fungi or their mycotoxins in food could be the use of MALDI-TOF mass spectrometry. MALDI-TOF offers a solution in the form of fast time detection (less than 10 min), and rapid sample preparation; it does not require chromatographic separation of metabolites and is not affected by buffers components or contaminants [13,14]. For example, MALDI-TOF MS has already been successfully applied for the identification of microscopic fungi such as *Aspergillus*, *Penicillium*, *Fusarium*, or *Trichoderma* and various yeasts (*Candida* spp.) from clinical samples [15,16,17]. In addition, MALDI-TOF MS was also used for the targeted determination of trichothecenes and their derivatives in barley [18], deoxynivalenol and nivalenol from barley malt [19], or for the detection of aflatoxins and citrinin [17].

Therefore, the aim of this study was to develop the rapid identification of AFB_1_ toxin and their producers by various direct and mainly indirect MALDI-TOF MS methodologies based on the discrimination of the entire mass spectrum.

## 2. Results and Discussion

### 2.1. Detection of AFB_1_

AFB_1_ is one of the most toxic of all mycotoxins [20] and is primarily produced as a secondary metabolite by the microscopic filamentous fungi *Aspergillus flavus* and *Aspergillus parasiticus* [21,22]. According to the International Agency for Research on Cancer, AFB_1_ belongs to the groups of the most toxic carcinogens classified in group I. Therefore, the importance of its detection and identification is significant in food and feed commodities. MALDI-TOF Mass Spectrometry is an important technique during the past decade, and it has been widely introduced and used as a diagnostic technique in many microbiology laboratories. In recent years, it has replaced many other procedures due to the high diagnostic accuracy, reliability, and speed of execution of a particular experiment or protocol [23].

The mass spectrum of the pure standard AFB_1_ was detected from 300 to 350 *m*/*z*, where it was identified as a pair of peaks in the spectrum with 313.109 *m*/*z* [M + H]^+^ and 335 059 *m*/*z* [M + Na]^+^ using MALDI-TOF MS Microflex LT (Figure 1). In both cases of aflatoxins solvent with and without the addition of NaCl, the spectrum quality was comparable but the signal intensity without the addition of NaCl was higher. However, as described by the team of Wang et al. [24] and Catharino et al. [25], the addition of NaCl enhanced the detection sensitivity of all aflatoxins to a LOD of 50 fmol. Therefore, this aspect has not been investigated in this study. The authors Catharino et al. [25] captured AFB_1_ as a standard mass spectrometry as a single peak in the spectrum with 335 *m*/*z* [M + Na]^+^, which was only a sodium adduct bound to AFB_1_. They did not capture pure AFB_1_ ion [M + H]^+^ in their experiment. In all cases, they describe measurements of the mass spectrum in the form of AFB_1_ with sodium adduct in the spectrum with 335 *m*/*z*. AFB_1_ as a pair of peaks with 313.19 *m*/*z* [M + H]^+^ and 335.22 *m*/*z* [M + Na]^+^ in the mass spectrum was detected by the team of Ventura et al. [26] using the single-quadrupole electrospray ionization tandem technique ESI-MS mass spectrometry. They used the same measurement in the positive ionization mode because measurement in the negative ionization mode shows a very low signal intensity.

### 2.2. Confirmation of AFB_1_ Producing Strains of Aspergillus flavus

This confirmatory part of the experiment was to detect AFB_1_ in *Aspergillus flavus* strains isolated from cereals, specifically barley and wheat. Isolated strains of *A. flavus* were cultured on an SDA medium and isolated with potentially occurring mycotoxins using 96% methanol. The mycotoxin was isolated from the mycelia and the nutrient medium around the mycelium. Potentially isolated mycotoxin in methanol was tested on a TLC plate. The resulting analysis by the TLC method confirmed the presence of AFB_1_ compared to the standard. In the last two decades, several methods have been developed for the detection of aflatoxin, including TLC (Thin Layer Chromatography) [27,28], enzyme-linked immunosorbent assay (ELISA) [29,30,31] or high-performance liquid chromatography (HPLC) [32,33]. Jangampalli et al. [34] also used the TLC method for initial qualitative detection and later adopted a more sophisticated method for qualitative and quantitative detection of AFB_1_ by HPLC. It was not necessary to analyze the amount of AFB_1_ produced in this experiment because the aim of the experiment was to rapidly diagnose *A. flavus*-producing strains only. Authors Císarová et al. [35] used the TLC method, where they detected the presence of AFB_1_ in producing strains of *A. flavus* and *A. parasiticus* after the application of various essential oils. They also did not use the quantitative HPLC method but only examined the production of aflatoxins; the aim was its complete elimination. The TLC method is a qualitative methodology, and therefore, was used as a verification method for MALDI-TOF mass spectrometry aflatoxin detection. Samples for MALDI-TOF MS analysis were obtained in three ways, scraping from TLC plates, isolating *A. flavus* mycelia, and isolating AFB_1_ from the medium around *A. flavus* colonies. A CHCA matrix (cinnamic acid) was used in the experiment. CHCA as a matrix has been used by many authors to investigate AFB_1_ in their studies [17,25,36]. The measuring range was performed from 150 to 500 *m*/*z*. This range was chosen to detect the monomer (190 *m*/*z*) and dimer (379 *m*/*z*) of the CHCA matrix, which served as a calibrant. Generally, the obtained mass spectrum of AFB_1_ by scraping from a TLC plate showed a relatively strong signal intensity (10^4^) in the case of the AFB_1_ standard with a spectrum of 335.237 *m*/*z* [M + Na]^+^ against AFB_1_ originating from producing strains.

#### 2.2.1. Direct AFB_1_ Detection from TLC Scrap

For analysis, the scraping samples of the mycotoxins from TLC plates by MALDI-TOF mass spectrometry were used as a positive sample only. There are a lot of proofs where many authors used different types of scraps from TLC plates for further analysis [37,38,39]. In our experiment, the mass spectrum of AFB_1_ was identified with a relative molecular weight of 335 *m*/*z* [M + Na]^+^ which represents AFB_1_ with sodium adduct (Figure 2). The pure AFB_1_ with a mass spectrum of 313 *m*/*z* [M + H]^+^ was not detected in any samples obtained by scraping, because sampling on the TLC plate was probably localized only to the site of the fluorescent spot containing AFB_1_ + Na (sodium adduct).

#### 2.2.2. Direct AFB_1_ Detection from *A. flavus* Mycelia

The extraction of aflatoxin from mycelia is a standard, widely used methodology that describes many authors from the past to the present [40,41,42]. In our study, the isolation of aflatoxin from mycelia was performed as standard for all samples and all samples were subjected to analysis, where the production and non-production of AFB_1_ in isolated strains of *A. flavus* was confirmed by MALDI-TOF mass spectrometry. Producing strains of *A. flavus* showed peaks in the mass spectrum with a relative molecular weight of 335 *m*/*z*. In the case of non-productive strains, the peak was absent (Figure 3). The standard was captured with high signal strength. The other samples showed lower signal intensity, but the obtained spectrum clearly confirmed the presence or absence of AFB_1_. A scrap from a TLC plate using pure AFB_1_ was used as the AFB_1_ standard.

#### 2.2.3. Direct AFB_1_ Detection from Medium around the *A. flavus*

It is well known that *Aspergillus flavus* and other fungi secrete their toxins directly into the medium [43,44,45]. Therefore, the same extraction method to isolate potentially present AFB_1_ from around the colonies in the medium was used and similar results were obtained. In the case of isolation from the medium, the signal intensity was comparable to the isolation technique where *A. flavus* fungal mycelium was used. In the producing strains, AFB_1_ was detected as a sodium adduct with a relative molecular weight of 335 *m*/*z*, and in the case of non-producing strains a 335 *m*/*z* peak was absent (Figure 4). The extraction of AFB_1_ using methanol was also performed by Catharino et al. [25], who isolated AFB_1_ from peanuts. Peanuts are a common nutrient medium for microscopic fungi of the genus *Aspergillus*. The isolated AFB_1_ was then tested by MALDI-TOF mass spectrometry and detected AFB_1_ as a peak in the mass spectrum with 335 *m*/*z* [M + Na]^+^. However, they state that they used ethylamine-CHCA cinnamic acid as a matrix. In none of the experiments did MALDI-TOF mass spectrometry identify a peak in the mass spectrum with 313 *m*/*z* [M + H]^+^, which represents the pure form of AFB_1_ as was the case with the detection of pure mycotoxin standards. In contrast, with HPLC-MS detection [46], it was possible to capture both peaks in the mass spectrum of AFB_1_ in silage maize samples: 313 *m*/*z* [M + H]^+^, 335 *m*/*z* [M + Na]^+^. However, the presence or absence of AFB_1_ can be detected by MALDI-TOF mass spectrometry in samples directly in the mycelium of the microscopic genus *Aspergillus*, or by extraction from the vicinity of the mycelium from the nutrient medium. As confirmed in previous studies, by preparing a suitable extraction method, it is possible to detect AFB_1_ in various samples, commodities, or foods by MALDI-TOF mass spectrometry, such as peanuts [25], barley [18], and others.

#### 2.2.4. Direct AFB_1_ Detection from Grain Samples

In contrast to previously described direct AFB_1_ detection methods using MALDI-TOF MS, the direct detection of AFB_1_ directly in grains was described in the literature only for a few mycotoxins such as fusarium mycotoxins [18,19,47] (fumonisins, nivalenol, zearalenon, aflatoxin G1 or deoxinivalenol) or alternaria mycotoxins [48] (alternariol, alternariol monomethyl ether, tentoxin).

Therefore, the direct detection of AFB_1_ directly from grains was included as one of the main methodological procedures in this experiment, because it speeds up the process of toxin identification.

In this experiment, the wheat and barley grain samples were infected in vitro. AFB_1_ was identified as a single peak with a relative weight of 335 *m*/*z*. For infected grains with an AFB_1_-producing strain of *A. flavus*, a positive result was obtained. Non-producing strains of *A. flavus* resulted in a negative (Figure 5). No differences were observed in the detection of AFB_1_ in infected wheat and barley; therefore, the mass spectra were presented as general in Figure 5. Elosta et al. [18] were able to detect deoxynivalenol and nivalenol using MALDI-TOF mass spectrometry, and also successfully tried to directly identify mycotoxins from barley. In addition to the different extraction methods used to isolate deoxynivalenol and nivalenol, they also used another type of matrix, namely 2,5-dihydroxybenzoic acid (DHB). DHBs, such as α-cyano-4-hydroxycinnamic acid (CHCAs), are suitable for the detection of small molecules such as mycotoxins, including aflatoxins [24,49]. In addition to the detection of fusarium toxins in barley, the authors Catharino et al. [25] researched the direct detection of aflatoxins in peanuts. They were able to use MALDI-TOF to identify multiple aflatoxins directly from samples without the need for cultivation and the subsequent procedures associated with aflatoxin extraction and detection.

### 2.3. Indirect Detection of Aflatoxin-Producing Strains of A. flavus by Mass Spectrum Discrimination

MALDI-TOF Mass Spectrometry was and is successfully used for discrimination at the genus and species level in various microorganisms such as bacteria [50,51,52], fungi [16,53,54,55], yeasts [56,57], algae [58], or other organisms such as freshwater fish [59], flee [60] and many more, which today forms the basis for identification in clinical microbiology especially [61,62,63]. MALDI-TOF MS is a rapid, accurate and very cost-effective tool not only for genus and species identification based on the entire spectrum, but it has been proven that it is also usable for different kinds of microorganism discrimination at the strain and subspecies level, which has grown rapidly in recent years and is slowly becoming part of routine identification. The literature contains many examples mainly from the bacterial realm and especially from the field of clinical microbiology, where knowledge of the subspecies is very important. From similar examples it is possible to mention serotyping for *Streptococcus pneumoniae* [64]; serotyping for *Legionella pneumophila* [65]; lineages of *Listeria monocytogenes* [66]; serotyping for *Salmonella enterica* [67]; or *Campylobacter jejuni* groups [68]. This research deals mainly with the issue of producing and non-producing strains of *A. flavus*. The literature describes many similar but not the same studies where MALDI-TOF MS was successfully used for discrimination between resistant and non-resistant bacterial strains (methicillin resistance, carbapenemase *Klebsiela pneumoniae* producers) [69,70,71], biofilm producers and non-producers [72,73], serovars [74,75,76], tetracycline producers [77], psychrotolerant *Bacillus cereus* group [78], *Vibrio cholerae* subspecies [79], for example.

Therefore, the main goal of this research was to discriminate the AFB_1_-producing and non-producing strains based on the entire mass spectrum. The entire mass spectra of producing and non-producing strains of *A. flavus* were confirmed by the TLC method and mass spectrometry. Fungal proteins were extracted by standard ethanol-formic acid-acetonitrile solution prior to the homogenization of *A. flavus* mycelia. CHCA cinnamic acid was used as a matrix. Standardized methodology (Bruker Daltonics, Germany, Bremen) using semi-automatic mass spectra collection was set. A 33-strain mass spectra of *A. flavus* in the range from 2000 to 20,000 *m*/*z* were collected. The obtained mass spectra were analyzed by flexAnalysis software (Bruker Daltonics, Germany, Bremen). Great emphasis was placed on the purity and quality of the mass spectra. After the initial *A. flavus* (producers and non-producers) mass spectra analysis and selection, they were transferred to the MALDI Biotyper OC software (Bruker Daltonics, Bremen, Germany) and used its statistical subroutines to create PCA dendrograms. Based on protein spectra statistical analysis the resulting PCA dendrogram divided the samples into two separate monasteries. Samples numbered from 1 to 16 represented AFB_1_-producing *Aspergillus flavus* strains. Samples numbered from 17 to 33 represented non-producing *A. flavus* strains. MALDI-TOF MS with appropriate software was able to discriminate producing and non-producing strains within one species into two significantly different monasteries (Figure 6). To date, no similar AFB_1_ mass spectra analysis at the strain level (producing and non-producing *Aspergillus flavus* strains) has been found. MALDI-TOF mass spectrometry was successfully used by Verwer et al. [80] on discrimination at the species level against *Aspergillus lentulus* and *Aspergillus fumigatus* only. All previously cited authors confirmed that MALDI-TOF mass spectrometry is a sensitive method that can distinguish very small variations in the mass spectrum and can be used to effectively discriminate between individual genera, species, strains, or subspecies and thus be used in microbial diagnostics.

#### *Aspergillus flavus* Entire Spectrum Observation

Detailed examination of protein spectra obtained from samples of producing and non-producing strains of *A. flavus* showed a significant difference in the individual spectra of producers and non-producers. After a deeper analysis of the mass spectra, differences mainly in the spectrum in the range from 4000 to 6800 *m*/*z* and 14,500 to 19,000 *m*/*z* were detected. Specifically, the main differences were found in the mass spectra of producing and non-producing *A. flavus* strains in molecules with a relative molecular weight of about 4040 and 4045 *m*/*z*, where these molecules occurred only in samples of non-producers (Figure 7).

Another significant difference was measured in the spectrum with a relative molecular weight of 6015 *m*/*z* and 6030 *m*/*z*, where ions of molecules (6015 *m*/*z*) occurred only in samples of aflatoxin-producing strains and ions of molecules (6030 *m*/*z*) only in samples of non-producing strains (Figure 8).

The spectrum of samples of producing and non-producing strains of *A. flavus* also differed in the spectrum from 7000 to 7200 *m*/*z*, where the spectrum shifted significantly towards lower values. The difference was also measured in the spectrum above 7400 *m*/*z* (Figure 9).

The most significant difference in the spectrum was in the range from 14,500 to 19,000 *m*/*z*, where there were two peaks with a relative molecular weight of about 15,500 *m*/*z* for three peaks from 18,000 to 18,500 *m*/*z*, which occurred only in samples of non-producing strains (Figure 10).

The analysis of protein mass spectra of AFB_1_-producing and non-producing *Aspergillus flavus* strains showed that non-producing *A. flavus* strains have a richer spectrum mainly in the lower molecular weight range, but there are also unique higher molecular weight mass spectra for non-producing *A. flavus* strains, which in the case of AFB_1_ producers do not occur.

The genus Aspergillus and its 12 species (*A. candidus, A. chevalieri*, *A. flavus*, *A. fumigatus*, *A. nidulans*, *A. niger, A. parasiticus*, *A. repens*, *A. sydowii*, *A. terreus*, *A. ustus* and *A. versicolor*) discriminated in the work of Hettick et al. [81] using MALDI-TOF mass spectrometry to the depth of species representation, found that the MALDI-TOF MS technique can be suitably used to diversify Aspergillus species. They analyzed the mass spectra between species of the genus Aspergillus in detail and did not pay attention to the differences in the mass spectra of different strains of the same species. A high-quality discrimination technique using MALDI-TOF MS was described in the study of Yoon et al. [82], who did not evaluate microscopic fungi, but *E. coli* and *K. pneumoniae* bacteria. Technically; however, the work is based on the same principles, where the authors were able to use mass spectra to differentiate individual strains of microorganisms within one species. Specifically, it was the differentiation of individual strains of *E. coli* and *K. pneumoniae*, where the authors were able to distinguish only slight differences between KPC-2, KPC-3 and KPC-4 carbapenemase-producing strains due to small variations in the mass spectrum. There are several references in the literature to MALDI-TOF discrimination of mass spectra, by which the authors wanted to diversify among various, sometimes very related samples, such as the discrimination of biofilm-producing and non-producing strains of Staphylococcus epidermidis [72], producing and low-producing strains of Candida parapsilosis [83], discrimination between plasmid and chromosomal polymyxin resistance in E. coli [84], or discrimination of β-lactamases in clinically important Enterobacteriaceae and Pseudomonas aeruginosa [85], discrimination between methicillin-resistant and susceptible species of Staphylococcus aureus [86]. The MALDI-TOF MS technique also interferes with other scientific disciplines, such as it is used in discriminating barley varieties based on detected β-hordeins [87,88], the discrimination of regional biotypes of Impatiens gland lifers using mass spectra obtained from the seeds of these plants [89], discrimination of the types of gum obtained from acacia [90] for example. Unfortunately, it was not possible to compare the results with other authors in the comparison of the mass spectra of producing and non-producing strains of *A. flavus*, because we have not yet found any relevant available literature regarding the mentioned issue. Therefore, it remains to be consulted and discussed in future studies on the results obtained for the difference in the mass spectra of aflatoxigenic and non-aflatoxigenic strains of Aspergillus flavus.

## 3. Materials and Methods

### 3.1. AFB_1_ Analysis

The first phase of the experiment was the detection of mycotoxin by MALDI-TOF mass spectrometry in pure form. In this experiment, the AFB_1_ mass spectrum was obtained from Sigma Aldrich (Munich, Germany) in high HPLC quality.

#### 3.1.1. Preparation of AFB_1_

AFB_1_ was solvated in methanol at a concentration of 0.1 mg/mL. After thorough homogenization, 1 μL of the solution was transferred to a MALDI-TOF stainless steel microplate into the appropriate targets. Every second repetition was prepared as a solution of mycotoxin at a concentration of 0.1 mg/mL in methanol with the addition of 10 M NaCl. After drying, the samples were overlaid with a matrix, consisting of CHCA (α-cyano-4-hydroxycinnamic acid) (Sigma Aldrich, Munich, Germany) solvated by a standard procedure in 250 µL of an organic solvent consisting of 50% acetonitrile (Sigma Aldrich, Munich, Germany), 47.5% deionized water and 2.5% trifluoroacetic acid (Sigma Aldrich, Munich, Germany). The matrix crystallized with the samples at room temperature.

#### 3.1.2. MALDI-TOF Detection of AFB_1_

The detection of AFB_1_ using MADLI-TOF MS was conducted by Catharino et al. [25] and Hleba et al. [17] with minor modifications. After the crystallization of the samples with matrix, mycotoxin mass spectra were measured on a MALDI-TOF Microflex LT mass spectrometer (Bruker Daltonics, Bremen, Germany) operating in a positive linear mode in cooperation with flexControl 3.4 software (Bruker Daltonics, Bremen, Germany). Spectra were measured in the range from 100 to 500 *m*/*z*. For the detection of small molecules of mycotoxin, the following device settings in the flexControl 3.0 software were performed: laser frequency at 60 Hz, laser attenuator in the range from 20 to 30%, detector gain at 1.0×, ion source 1 at 18 kV, ion source 2 at 15 kV, lens at 6 kV, pulse ion extraction at 0 ns. Mass spectra were collected in a randomized manner for 500 shots per sample. The CHCA matrix (α-cyano-4-hydroxycinnamic acid) (Sigma Aldrich, Munich, Germany) in pure form was used as a calibrant and a negative control at the same time.

#### 3.1.3. Analysis of Mass Spectra

Spectral analysis was performed in flexAnalysis 3.4 software (Bruker Daltonics, Bremen, Germany). The detection of mycotoxin spectra was performed based on a centroid detection algorithm with signal and background noise thresholds of 1, relative noise 0%, minimum noise 0, peak width 0.2 *m*/*z*, peak height at 80% and Tophat by basic subtraction, without the need for a smoothing algorithm. The theoretical molecular weight with peaks for the AFB_1_ was compared with the detected spectrum and peaks with a maximum deviation of 0.5 *m*/*z*. Analyzed mass spectra were presented as an average of 500 shots for each tested sample of AFB_1_-producing and non-producing *A. flavus* strains (33 strains) and the pure form of AFB1 was presented as an average of 500 shots in triplicate.

### 3.2. MALDI-TOF Direct Detection of AFB_1_ in Producing Strains Aspergillus flavus

#### 3.2.1. Microorganisms, Cultivation, and Identification

*Aspergillus flavus* strains (33 isolates) were isolated from wheat and barley samples from the surface of individual grains. The initial culture was cultivated on DRBC agar (HiMedia, Mumbai, India) at 25 °C for 5–7 days. After cultivation, the morphological identification of macromorphological and micromorphological features of *Aspergillus flavus* based on identification keys [91,92,93,94] was conducted. Strains identified as *Aspergillus flavus* were inoculated onto Saboraud dextrose agar (SDA) (HiMedia, Mumbai, India), where they were cultured at 25 °C for 5–7 days.

#### 3.2.2. MALDI-TOF MS *A. flavus* Identification

Fungal isolates were sub-cultured on SDA agar (HiMedia, Mumbai, India) plates at 25 °C for 5–7 days until sufficient fungal growth was observed. Appropriate fungal mycelia were transferred into the Eppendorf tube containing 300 µL deionized water and 900 µL of 100% ethanol (Sigma Aldrich, Munich, Germany) and vortexed. After disintegration, the fungal mycelia solutions were centrifuged at 12,000× *g* rpm and the ethanol/water mixture was removed by pipetting. The rest solution was air-dried at room temperature. Eppendorf tubes with fungal mycelia were filled with 50 µL of formic acid (Sigma Aldrich, Munich, Germany) and vortexed with glass balls for thorough mycelia disintegration. Suspensions were mixed with an equal volume of acetonitrile (Sigma Aldrich, Munich, Germany) and vortexed. Acquired samples were 12,000× *g* rpm for 2 min and 1 µL of supernatants were placed into a MALDI steel plate (MSP 96 target plate, Bruker Daltonik, Bremen, Germany) and overlaid with a matrix (α-cyano-4-hydroxycinnamic acid solution, Sigma Aldrich, Munich, Germany) (1 µL) prepared by the Bruker Daltonik protocol. The samples on the MALDI plate were allowed to dry at room temperature. Following, the proteins were extracted using a formic acid/ethanol (Sigma Aldrich, Munich, Germany) procedure, according to the Bruker Daltonik standard protocol.

#### 3.2.3. Verification of AFB_1_ Producers by TLC Method

The TLC qualitative method (as MALDI-TOF MS) for the detection of aflatoxin-producing strains was chosen as a confirmatory model of AFB_1_-producing *Aspergillus flavus* isolates. The biomass of the microscopic fungi together with a 1 cm^2^ section of agar was transferred to an Eppendorf tube containing 500 µL of extractant (96% methanol) (Centralchem, Bratislava, Slovakia). The samples were homogenized by vortex and then 14,000× *g* rpm. A 30–50 µL sample was pipetted from the obtained supernatant and transferred to an Alugram Sil G/Plaque finies CCM alugram Sil G chromatographic plate (Macheres-Nagel, Dtiren, Germany). The pure AFB_1_ standard was prepared in the same solvent and applied together with the samples onto a TLC plate. The mobile phase was formed by formic acid complex:ethyl acetate:toluene in a ratio of 1:3:5. Visualization of AFB_1_ was performed under ultraviolet radiation at a wavelength of 365 nm as the blue spot with an R_F_ value of 0.56 [95]. Unused sample material for the isolation of AFB_1_ stored in Eppendorf tubes was used after 14,000× *g* rpm for the following experiments using MALDI-TOF mass spectrometry. As a control, positive test samples to produce AFB_1_ detected by the TLC method by scraping from positively identified spots on the TLC plate were used. Samples obtained by scraping from TLC plates were re-solvated in 96% methanol, homogenized by vortex and 14,000× *g* rpm until a pellet formed from silica gel residues originating from the TLC plate [96]. From each sample, fungal biomass was scraped from a TLC plate and extracted with medium. One microliter was pipetted and transferred on a MALDI-TOF stainless steel plate to the appropriate targets. After drying at room temperature CHCA matrix was diluted in an organic solvent (previously described) overlaid samples and allowed to crystallize freely at room temperature. AFB_1_ solvated in 96% methanol at a concentration of 0.1 mg/mL was used as a standard. As a negative control, a non-producing strain identified during the TLC method was used.

#### 3.2.4. Mass Spectrometry Methodology

The mass spectra were obtained on a MALDI-TOF mass spectrometer Microflex LT operating in a positive linear mode with a range of 100 to 500 *m*/*z* of the observed spectrum. Other parameters of the instrument setup were the same as for obtaining the mass spectra of the pure form of mycotoxin in the previous chapter. FlexAnalysis 3.4 software was used to analyze the mass spectra obtained in the MALDI-TOF mass spectrometry process. All software settings for mass spectral analyses were the same as for mass spectral analysis of pure forms of mycotoxin, described in the previous chapter.

### 3.3. MALDI-TOF MS Detection AFB_1_ in Grain Samples Directly

Wheat and barley grains were soaked in hot water to increase humidity. After soaking the grains were infected by producing and non-producing AFB_1_  *Aspergillus flavus* strains. Grains were incubated with fungi at 25 °C for 3 days. Subsequently, grains were surface sterilized with 5% sodium hypochlorite for 5 min, followed by washing in sterile distilled water and drying with an air flow drier. Dried grains were homogenized and dissolved into the same solution as mycelia and medium previously described. The resulting suspension was homogenized at 14,000× *g* rpm until pellet formation; 1 µL of the supernatant was transferred to a MALDI-TOF stainless steel plate. CHCA (as matrix) was used.

### 3.4. MALDI-TOF Detection of AFB_1_ Producing Strains of Aspergillus flavus Based on Comparison of Entire Mass Spectra (Fingerperint Method)

#### 3.4.1. Proteins Isolation and MALDI-TOF Protocol

In the fingerprint method of comparing the entire mass spectra of producing and non-producing *Aspergillus flavus* strains, the mass spectra were obtained by acetonitrile/water/formic acid protein isolation procedure from the biomass of microscopic fungi of *Aspergillus flavus*. Isolation of intact proteins was achieved by homogenization of samples using glass beads in Eppendorf tubes in a Beads bug cell homogenizer (BenchMark Scientific, Sayreville, NJ, USA) due to membrane disintegration. After homogenizing the AFB_1_ samples of the producing and non-producing strains, samples were centrifuged at 14,000× *g* rpm until pellet formation. From the resulting supernatant, 1 μL was transferred to a MALDI-TOF microplate, and after drying, covered with CHCA matrix dissolved in the same organic solvent as in the previous experiment. The mass spectra were obtained on a MALDI-TOF Microflex LT mass spectrometer in a positive linear mode in the range of the observed mass spectrum from 2000 to 20,000 *m*/*z*. Using the flexControl 3.4 control software, the instrument settings were left in a standardized mode for collecting mass spectra in the standard microorganism identification procedure according to Bruker Daltonics, Germany. Obtained mass spectra were of high quality with high reproducibility of the results. The rules of the achieved score in identification were used as a standard in the work. An obtained score higher than 2 was considered because of the identification of the species, a score of 1.7–2 for genus identification and a score lower than 1.7 for unreliable identification. In this experiment scores higher than 2 were used for identification only. Reference strain *Escherichia coli* DH5alpha (ref. Strain 255343, Bruker Daltonics, Bremen, Germany) was used in this experiment to check the purity of the spectrum. In the initial identification of *Aspergillus flavus*, reference mass spectra of *Aspergillus flavus* strains stored in the Bruker Daltonics database (Germany), which is part of the MALDI Biotyper identification software, were used for comparison and identification too.

#### 3.4.2. Creating an *Aspergillus flavus* Local Database

Similar to authors Loucif et al. [97] and Hleba et al. [77], an *Aspergillus flavus* local database consisting of protein spectra obtained from AFB_1_ producing and non-producing *Aspergillus flavus* strains identified in a standardized identification process on a MALDI-TOF mass spectrometer was created. After precise control of the mass spectra in the flexAnalysis ver. 3.0 and identification in MALDI Biotyper OC ver. 3.1, MSP creation (Mass Spectrum Projection) using the Biotyper MSP methodology set by Bruker Daltonis was conducted.

#### 3.4.3. Verification of Reproducibility

By retrospectively verifying the reproducibility of mass spectra results obtained from *Aspergillus flavus* samples, the local *Aspergillus flavus* database and randomly detected mass spectra of randomly selected *A. flavus* samples were compared in MALDI Biotyper OC ver. 3.1.

## 4. Conclusions

Based on all confirmatory methods of direct AFB_1_ detection by MALDI-TOF mass spectrometry (direct detection AFB_1_ from mycelium of producing strain of *A. flavus*, from medium around the growth of a producing strain of *A. flavus*, from the TLC scraps and direct detection from the infected grains by a producing strain of *A. flavus*) and the TLC method, it is possible to confirm that MALDI-TOF mass spectrometry can be successfully used to indirectly detect AFB_1_ or discrimination between producing and non-producing strains of *A. flavus.* The methodology was based on the entire mass spectrum discrimination ranging from 2000 to 20,000 Da. However, there is a need to create a local entire mass spectra database of producing and non-producing strains of *A. flavus*. If there is a local database of producers and non-producers, it will be included in the classic identification scheme using MALDI-TOF mass spectrometry. It is possible to distinguish aflatoxigenic strains of *A. flavus* right at the beginning of identification, and immediately after cultivation. The main advantage is cost reduction and ultra-fast identification of aflatoxigenic strains of *A. flavus*.

## Figures and Tables

**Figure 1 molecules-27-07861-f001:**
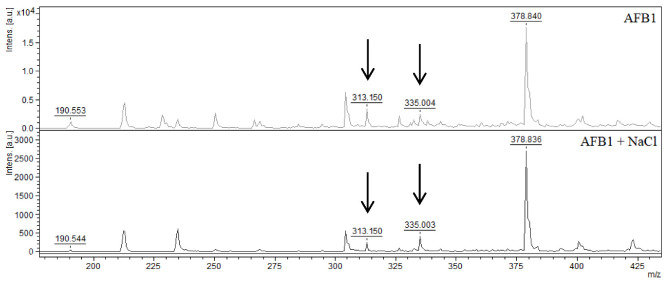
MALDI-TOF mass spectra of AFB_1_ presented as tandem peaks with mass spectrum of 313.150 *m*/*z* [M + H]^+^ and 335.004/335.003 *m*/*z* [M + Na]^+^. Peaks are indicated with arrows.

**Figure 2 molecules-27-07861-f002:**
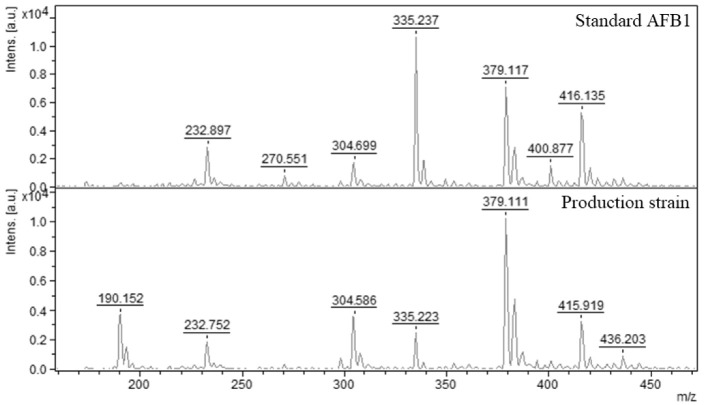
MALDI-TOF mass spectra of AFB_1_ obtained by scraping from TLC plate. Standard AFB_1_ (335.237 *m*/*z*) and producing strain of *A. flavus* (335.223 *m*/*z*).

**Figure 3 molecules-27-07861-f003:**
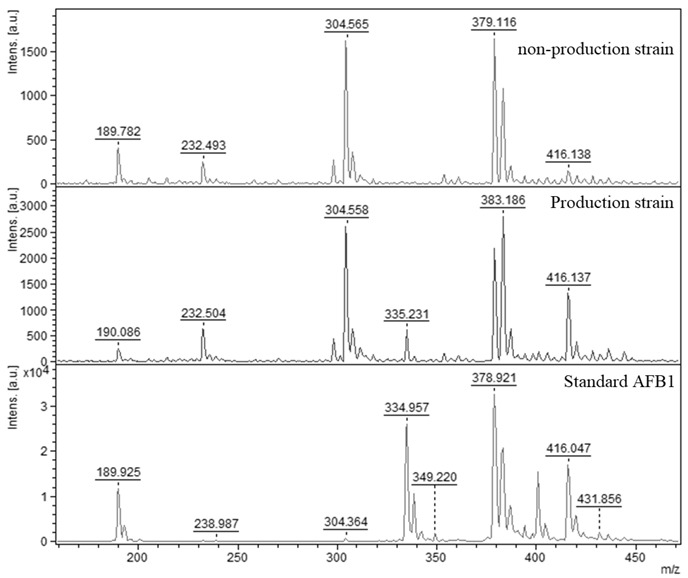
MALDI-TOF mass spectra of AFB_1_ obtained by isolation from *Aspergillus flavus* mycelia. Standard AFB_1_ (334.957 *m*/*z*), non-producing strain (peak absent) and producing strain with the peak (335.231 *m*/*z*).

**Figure 4 molecules-27-07861-f004:**
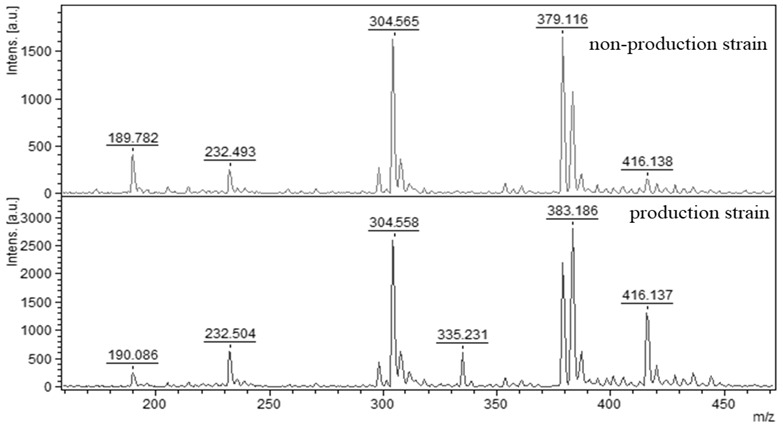
MALDI-TOF mass spectra of AFB_1_ obtained by isolation from medium around the *Aspergillus flavus* mycelia. Producing strain (335.231 *m*/*z*) and non-producing strain (without peak of alfatoxin B_1_).

**Figure 5 molecules-27-07861-f005:**
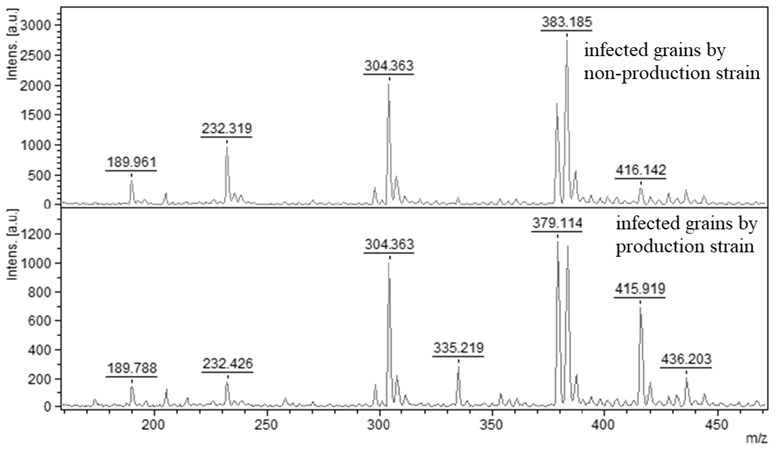
MALDI-TOF mass spectra of AFB_1_ isolated directly from wheat and barley grains infected by producing strain (335.219 *m*/*z*) and non-producing strain of *A. flavus* (AFB_1_ peak absent).

**Figure 6 molecules-27-07861-f006:**
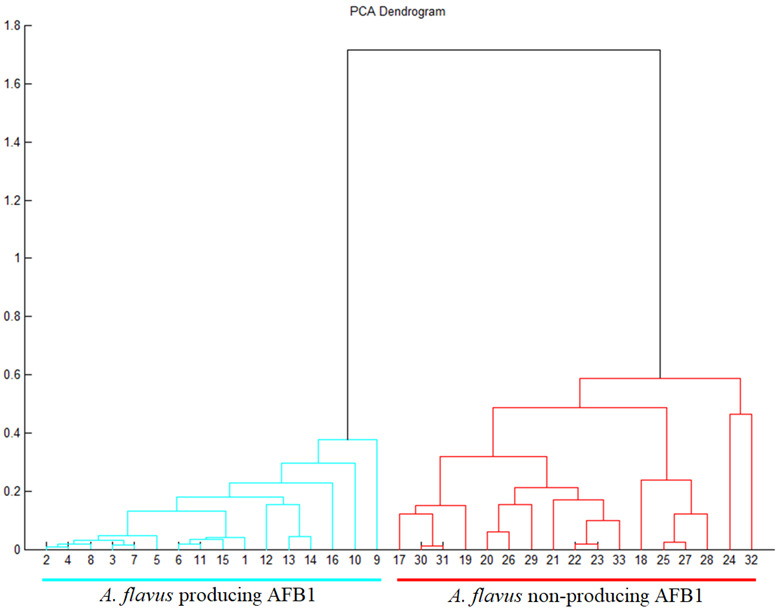
PCA (Principal Component Analysis) dendrogram presented relatedness between *Aspergillus flavus* producing AFB_1_ and non-producing AFB_1_ strains. Statistical evaluation divided producing and non-producing strains onto two clusters marked in different colors.

**Figure 7 molecules-27-07861-f007:**
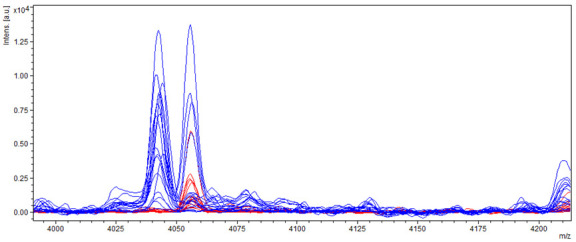
Differences in mass spectra detected in range of about 4040 and 4045 *m*/*z* from AFB_1_ producing and non-producing strain of *A. flavus*, where blue spectra showed non-producers and red spectra producers.

**Figure 8 molecules-27-07861-f008:**
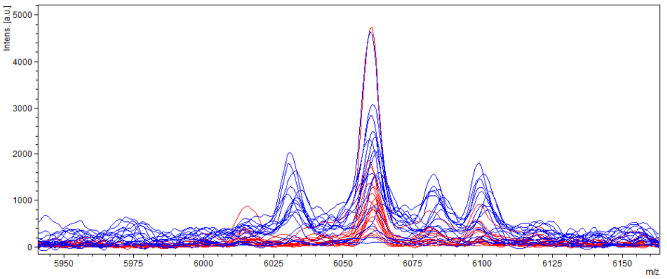
Differences in mass spectra detected in range of about 6015 and 6030 *m*/*z* from AFB_1_ producing and non-producing strain of *A. flavus*, where blue spectra showed non-producers and red spectra producers.

**Figure 9 molecules-27-07861-f009:**
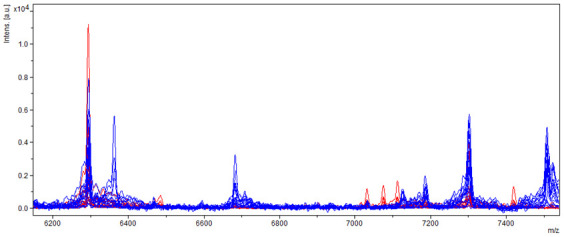
Differences in mass spectra detected in range of about 7000 and 7200 *m*/*z* from AFB_1_ producing and non-producing strain of *A. flavus*, where blue spectra showed non-producers and red spectra producers.

**Figure 10 molecules-27-07861-f010:**
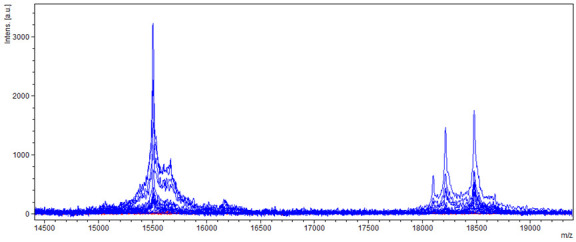
Differences in mass spectra detected in the range from 14,500 to 19,000 *m*/*z* from aAFB_1_ producing and non-producing strain of *A. flavus*, where blue spectra showed non-producers and red spectra producers.

## Data Availability

Not applicable.

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
