# Peer review of "Use of MALDI-TOF MS to Discriminate between Aflatoxin B1-Producing and Non-Producing Strains of Aspergillus flavus"

_molecules, 2022, doi:10.3390/molecules27227861_

Round 1

Reviewer 1 Report

This is a very practical manuscript dealing with a fast method (MALDI-TOF MS) to identify aflatoxin B1-producing strains of Aspergillus flavus. I only have few minor comments to be addressed before publication can be recommended:

1.       Please, provide, at least, one reference supporting the use of your analytical methods in previous successful studies.

2.       I would like to see some basic information about how much time is save by using this new identification method as compared to the existing ones, and also some basic information about the comparative costs of the proposed and existing methods. This information is basic for potential users to decide whether this is a practical and real option for them or not.

3.       Figures 1 to 5, please, indicate the number of replicates used; since it is not possible to represent any statistical information in these figures, at least, indicate the replication of the experiment.

Author Response

Dear reviewer,

thank you very much for your suggestion in this review process. 

This is a very practical manuscript dealing with a fast method (MALDI-TOF MS) to identify aflatoxin B1-producing strains of Aspergillus flavus. I only have few minor comments to be addressed before publication can be recommended:

Point 1: Please, provide, at least, one reference supporting the use of your analytical methods in previous successful studies.

Response 1: Dear reviewer, thank you very much for your comments. We added some previously published studies to methodology as citations. Explanation: Preparation of AFB1 was done by standard methods in methanol. Detection of AFB1 by MALDI-TOF MS we prepared by Cathrino et al., and Hleba et al. – these authors were added to article. Spectral analysis was done in software with setup to specific spectrum. Identification of A. flavus – macro a micromorphological – citations were added. MALDI identification was preformed by standard Bruker procedure. Verification AFB1 by TLC and extraction procedure – citation was added. Mass spectrum was obtained by standard Bruker procedure. Isolation of AFB1 from grains were the same as in TLC methodology – it was cited. Fingerprinting method was done by standard Bruker procedure for mass spectra acquiring. Local database creating was done by Loucif et al., and Hleba et al. Its mean that all methodology where the spectra were acquired is based on standard Bruker procedure. Other methodologies were cited. Thank you for comment for improving our article.

Point 2: I would like to see some basic information about how much time is save by using this new identification method as compared to the existing ones, and also some basic information about the comparative costs of the proposed and existing methods. This information is basic for potential users to decide whether this is a practical and real option for them or not.

Response 2: Dear reviewer, we cannot accurately estimate the time or cost of the difference between our proposed alternative way of identification and the other methods, because we only used the methods mentioned in the paper. In addition, each chemical purchased may have a different price from different manufacturers, and prices also change over time, so it is impossible to estimate exactly how much identification would cost. It is also difficult to estimate the time of the identification, because everyone has different abilities and speed of work. However, what we could and did mention in the work is the general evaluation of time and finances, which we stated at the conclusion of the work, where we mention that if the database of mass spectra is completed, it is possible to identify the production strain of A. flavus immediately after cultivation, and thus all other confirmatory methods used before.

Point 3: Figures 1 to 5, please, indicate the number of replicates used; since it is not possible to represent any statistical information in these figures, at least, indicate the replication of the experiment.

Response 3: Dear reviewer, thank you for suggestion. We added this information to section 3.1.3 Analysis of mass spectra. Explanation: All our samples including all 33 strains of A. flavus, 16 aflatoxinogenic and 17 non-aflatoxinogenic strains, 16+17 TLC scraps, 16+17 medium, 16+17 mycelium, 16+17 infected grains and pure mycotoxin AFB1 in triplicate x 500 shots for 1 sample we done. For example Figure 4 was created as average of 16 aflatoxinogenic and 17 non-aflatoxinogenic samples collected by 500 laser shots.

All corrections are showed into the text directly in yellow.

Dear reviewer,

Thank you very much for suggestions in this article. We hope that we have added and explained all of yours suggestions or reminders and we hope that you are satisfy with our explanations.

Thank you again

Authors

Reviewer 2 Report

This work examined a simple, sensitive, and reliable analytical method for rapidly identification of producing and non-producing strains of A. flavus. To accomplish the goal, direct AFB1 identification by scraping from TLC plates, A. flavus mycelium, nutrient media around A. flavus growth, and finally direct AFB1 identification from infected wheat and barley grains had to be done. In this experiment MALDI-TOF mass spectrometry as main technology was used. The results are relevant but some considerations need to be clarified. Moderate English changes required.

- AFB1 was identified by A. flavus mycelium and nutrient media around A. flavus growth. How to completely separate the mycelium and nutrient media?

- 68:The reference [17] cited in the paper also detects AFB1 using MALDI-TOF mass spectrometry, please elaborate the difference and innovations in the discussion.

- samplings were all on the TLC plate, Figure 3: Standard AFB1 (334.957 m/z) from TLC plates, but Figure 2: producing strain of A. flavus (335.223 m/z) from TLC plates, why the m/z was different?

- Almost all of the methods sections have a common serious error, missing references, please add!

- 34, 155-156:AFB1? The first time it appears with the full name, all of “aflatoxin B1” should be modified to the abbreviated name “AFB1” later, please check the whole text.

-. 190. 210: MALDI-TOF? The writing style should be uniform throughout the text.

- 231: lineages of Listeria monocytogenes [66];

- 268: A. flavus

- 146. 268. 310-311, 318, 321, 327, 328, 331, 337, 340, 386:species not in italic. A large number of strains appear not in italics, please check the whole text.

- 406:cm2

- 424: reference?

- 426: 0.1 mg.ml-1

Author Response

Dear reviewer,

thank you very much for your suggestion in this review process.

Response to Reviewer 2 Comments

This work examined a simple, sensitive, and reliable analytical method for rapidly identification of producing and non-producing strains of A. flavus. To accomplish the goal, direct AFB1 identification by scraping from TLC plates, A. flavus mycelium, nutrient media around A. flavus growth, and finally direct AFB1 identification from infected wheat and barley grains had to be done. In this experiment MALDI-TOF mass spectrometry as main technology was used. The results are relevant but some considerations need to be clarified. Moderate English changes required.

Point 1:  AFB1 was identified by A. flavus mycelium and nutrient media around A. flavus growth. How to completely separate the mycelium and nutrient media?

Response 1: Response: the mycelium was collected from the surface of the plates and the nutrient medium was taken from the places in close proximity to the growth of the mycelium, where the mycelium did not yet interfere (these places were identified and viewed under a microscope, marked and separated)

Point 2: The reference [17] cited in the paper also detects AFB1 using MALDI-TOF mass spectrometry, please elaborate the difference and innovations in the discussion.

Response 2: Reference number 17 is work we have done in the past. However, the work deals only with the detection of pure forms of mycotoxins. The aim of this study was to determine which of the applied mycotoxins could be detected by MALDI TOF MS. Therefore, we did not include this information directly in the discussion. However, we listed it in the material and methodology section because we built the AFB1 identification section on this work. Thanks for the suggestion.

Point 3: samplings were all on the TLC plate, Figure 3: Standard AFB1 (334.957 m/z) from TLC plates, but Figure 2: producing strain of A. flavus (335.223 m/z) from TLC plates, why the m/z was different?

Response 3: When analyzing mass spectra, small deviations occur regularly, it always depends on the current situation of various variables in the device (laser, attenuator, vacuum, etc.), therefore small differences are always visible in m/z. In section 3.1.3, at the end of the paragraph, we also mention a possible deviation of 0.5 m/z.

Point 4: Almost all of the methods sections have a common serious error, missing references, please add!

Response 4: Dear reviewer, thank you very much for your comments. We added some previously published studies to methodology as citation. Explanation: Preparation of AFB1 was done by standard methods in methanol. Detection of AFB1 by MALDI-TOF MS we prepared by Cathrino et al., and Hleba et al. – these authors were added to article. Spectral analysis was done in software with setup to specific spectrum. Identification of A. flavus – macro a micromorphological – citations were added. MALDI identification was preformed by standard Bruker procedure. Verification AFB1 by TLC and extraction procedure – citation was added. Mass spectrum was obtained by standard Bruker procedure. Isolation of AFB1 from grains were the same as in TLC methodology – cited. Fingerprinting method was done by standard Bruker procedure for mass spectra acquiring. Local database creating was done by Loucif et al., and Hleba et al. Its mean that all methodology where the spectra were acquired is based on standard Bruker procedure. Other methodologies were cited. Thank you for comment for improving our article. 

Point 5: 34, 155-156:AFB1? The first time it appears with the full name, all of “aflatoxin B1” should be modified to the abbreviated name “AFB1” later, please check the whole text.

Response 5: Thank you very much for suggestion. We revised and modified AFB1 into the whole text.

Point 6:. 190. 210: MALDI-TOF? The writing style should be uniform throughout the text.

Response 6: We uniformed all MALDI-TOF short into the whole text

Point 7: 231: lineages of Listeria monocytogenes [66];

Response 7: revised

Point 8: 268: A. flavus

Response 8: revised

Point 9: 146. 268. 310-311, 318, 321, 327, 328, 331, 337, 340, 386:species not in italic. A large number of strains appear not in italics, please check the whole text.

Response 9: All name of microorganism are in italics

Point 10: cm2

Response 10: revised

Point 11: reference?

Response 11: citation was added.

Point 12: 0.1 mg.ml-1

Response 12: revised – units was uniformed into the whole text to mg/mL

All corrections are showed into the text directly in yellow.

Authors letter to reviewers:

Dear reviewers,

Thank you very much for suggestions in this article. We hope that we have added and explained all of yours suggestions or reminders and we hope that you are satisfy with our explanations.

Thank you again

Authors